# Effects of Licorice Extract Supplementation on Feed Intake, Digestion, Rumen Function, Blood Indices and Live Weight Gain of Karakul Sheep

**DOI:** 10.3390/ani9050279

**Published:** 2019-05-24

**Authors:** Xuefeng Guo, Long Cheng, Junfeng Liu, Sujiang Zhang, Xuezhao Sun, Omar Al-Marashdeh

**Affiliations:** 1College of Animal Science, Tarim University, Alar, Xinjiang 843300, China; zsjdky@126.com; 2Key Laboratory of Tarim Animal Husbandry Science and Technology of Xinjiang Production and Construction Group, Alar, Xinjiang 843300, China; 3Faculty of Veterinary & Agricultural Sciences, Dookie Campus, The University of Melbourne, Victoria 3647, Australia; long.cheng@unimelb.edu.au; 4The Center for Ruminant Precision Nutrition and Smart Farming, Jilin Agricultural Science and Technology University, Jilin 132109, China; xuezhaos@hotmail.com; 5Faculty of Agriculture and Life Sciences, Lincoln University, Canterbury 85084, New Zealand; omar.al-marashdeh@lincoln.ac.nz

**Keywords:** feed additive, bioactive compound, fermentation, antioxidant, immunity indices

## Abstract

**Simple Summary:**

The aim of this study was to investigate the effects of licorice extract supplementation on feed intake and digestibility, rumen function, blood indices and live weight gain of Karakul sheep. The results indicated that licorice extract supplementation in the diet at 4.5% of dry matter level had limited impact on rumen function but improved blood immunoglobulin and anti-oxidative status of Karakul sheep, without impairment of feed conversion efficiency or live weight gain.

**Abstract:**

This study was designed to investigate the effects of licorice extract supplemented to a cottonseed hull-based diet on rumen function, blood indices and growth of Karakul sheep. Twelve rumen-fistulated 1.5-year-old sheep were blocked in pairs by live weight. Sheep within pairs were randomly allocated to feed either on a cottonseed hull basal diet (control group) or on a basal diet containing 4.5% of licorice extract on dry matter (DM) basis (supplemented group). Sheep were housed individually and fed for 60 days, with a 15-day adaptation period and a 45-day measurement period. Feed intake and live weight gain were quantified. Rumen and blood samples were taken during the measurement period. The DM intake was lower for the supplemented group than for the control group. The mean ruminal concentrations of total volatile fatty acid (VFA) and acetate and the ratio of acetate to propionate were lower, while the levels of propionate and butyrate were higher for the supplemented group than for the control group. Average daily live weight gain, digestible energy intake, apparent nutrient digestibility and feed conversion efficiency did not differ between the two treatments. The serum concentrations of immunoglobulin A and G were 2.1 and 1.8 times greater, and total antioxidant and superoxide dismutase increased by 1.8 and 1.2 times in the supplemented group compared with the control group. These results indicated that licorice extract supplementation in the diet at 4.5% of DM had a limited impact on rumen function but improved blood immunoglobulin and anti-oxidative status of Karakul sheep, without impairment of feed conversion efficiency or live weight gain.

## 1. Introduction

The use of antibiotics as feed additives for livestock production has been restricted worldwide over the last decade. The restriction is due to the concerns over the spread of resistant pathological bacteria and the residues of antibiotics in animal products [1,2]. This has raised the interest in alternative phytogenic feed additives to improve livestock health and production [3,4]. One potential option is the use of licorice (*Glycyrrhiza uralensis*) root extract as a feed additive, which has been investigated mainly in monogastrics over the last 15 years [5,6,7] and scarcely in ruminants. Recently, Zhang et al. (2015) showed that supplementation of a 0.4% licorice root extract (16.4% total flavonoids content) to sheep diet resulted in a greater antioxidant capacity of meat compared with the non-supplemented group [8]. Furthermore, Sajjadi et al. (2014) demonstrated that supplementation with 10 g day^−1^ of licorice root extract (equivalent to 0.56% of the diet) increased total immunoglobulin (Ig) concentration and, hence, improved the immunological status of Holstein heifer calves [9]. On the other hand, Kim et al. (2013) showed that supplementation of the diet with 0.5% licorice (no licorice chemical composition reported) had no effect on dry matter intake (DMI) and average daily gain (ADG) of Hanwoo steers [10]. Although these studies suggest potential positive effects of licorice extract supplementation on antioxidant and immunological status of animals, information is lacking on the maximal level of licorice extract that is required to optimise these potential benefits while maintaining rumen function, DMI and subsequent ADG.

In a previous in vitro study, we investigated the effects of adding a licorice extract (22% glycyrrhizin content) to the feed at different levels, i.e., 0.0%, 2.25%, 4.5%, 13.5%, 22.5% and 31.5% of dry matter (DM), on sheep rumen fermentation characteristics [11]. The results showed that such licorice extract had limited effect on rumen function when the level of supplementation was less than 4.5% of DM in the feed. However, when more than 4.5% licorice extract was added to the fermentation substrate, both ruminal gas and volatile fatty acid production was reduced, suggesting that rumen fermentation was impaired. While the study suggested that licorice supplementation at 4.5% of DM can maintain in vitro rumen fermentation, there is still a lack of in vivo information on the effects of licorice supplementation on rumen function, DMI and ADG. Thus, the hypothesis of this study was that adding 4.5% (DM basis) of licorice extract (22% glycyrrhizin content) to the diet would have little effect on DMI, digestion, rumen function and ADG, but would improve immunological and antioxidant parameters in the blood of sheep. The choice of the 4.5% (on DM basis) licorice extract supplementation level in this study was to (1) confirm previous in vitro results through this in vivo sheep study and (2) explore the effect of the highest known licorice extract supplementation level (i.e., 4.5% on DM basis, based on the results of previous literature) on rumen function, digestion, antioxidant and immunological status and sheep growth.

## 2. Materials and Methods

### 2.1. Experimental Site and Design

The experiment was conducted at the Animal Research Station, Tarim University (East longitude: 81°17′35.37″; North latitude: 40°32′28.99″), China, from 1 December 2009 to 30 January 2010. Twelve Karakul wethers with rumen fistulae installed when they were 1 year old, aged 1.5 ± 0.25 years (mean ± SD), were blocked in pairs by their initial live weights (29.4 ± 0.3 kg). Wethers within pairs were randomly assigned to one of two treatment groups: (1) cottonseed hull-based diet, containing 70% cotton seed hulls, 18.2% maize, 6.6% wheat bran, 3.0% soybean meal, 0.36% urea, 0.13% lime stone, 0.28% CaHPO_4_, 1% NaCl, 0.33% Na_2_SO_4_ and 0.1% vitamin premix (Vitamin A, D3 and E) on DM basis (Control group; *n* = 6) and (2) cottonseed hulls-based diet + 4.5% licorice extract on DM basis (Supplemented group; *n* = 6). The fistulae were made of silica gel with an internal diameter of 3.0 cm and length of 6.0 cm (Chinese Agriculture University, Beijing, China). All sheep were housed individually in metabolic cages (1.2 m × 1.5 m) and fed the experimental diet individually ad libitum twice a day at 8:00 a.m. and 8:00 p.m.. Sheep had free access to clean water throughout the experiment. The experiment was conducted over a period of 60 days, including 15 days for adaptation and 45 days for measurements.

The basal diet was formulated using typical feed ingredients sourced locally to meet animal requirements for 100 g of ADG, according to the Chinese Feeding Standard of Meat-Producing Sheep and Goats [12]. A 50% licorice root ethanol extract was purchased (Xinjiang Talimu Agriculture Development Co., Ltd., Xinjiang, China) and administered to sheep at 4.5% of DM in the diet, which had a *glycyrrhizin* content equivalent to 1.0% of DM. The licorice extract content of *glycyrrhizin* was determined using HPLC [13].

### 2.2. Animal Ethical Code

This study was approved by the Tarim University Animal Care and Use Committee. All procedures used in this study are in agreement with the Chinese Guidelines for the Care and Use of Animals for Research (GB 14925, 2001) [14]. 

### 2.3. Measurements and Analysis

#### Dry Matter Intake and Feed Analysis

The DMI was measured daily for individual sheep by subtracting the feed refused from the feed offered and then correcting the result for DM content.

Approximately 100 g of feed was sampled twice a day before each meal. Weekly samples were bulked, and a 10% subsample was oven-dried at 65 °C for 48 h to determine the DM content. The dried samples were ground through a 1 mm sieve (Cyclotec 1093; FOSS Ltd., Hillerød, Denmark) prior to chemical analysis. Gross energy was determined using the oxygen bomb calorimeter (Digimatic Oxygen Bomb Calorimeter HR-15; Changsha Changxing High Grade Educational Equipment Development Co., Ltd., Changsha, China). Nitrogen (N) content was determined using the Kjeltec 8400 Analyzer Unit (Kjeltec 8400; FOSS Ltd., Hillerød, Denmark) and cupric sulfate and vitriolate of soda as catalyzers. Crude protein (CP) was calculated according to the equation of CP% = N% × 6.25. Neutral detergent fiber (NDF) and acid detergent fiber (ADF) were analyzed using the Fibertec M6 Fiber Analyzer (FOSS Ltd., Hillerød, Denmark). In NDF analysis, 3.0% lauryl sodium sulfate was used as a detergent, and 0.5 g of anhydrous sodium sulfate per 1.0 g sample was added. NDF and ADF were expressed ash-free [15].

### 2.4. Feces Sampling, Digestible Energy Intake and Apparent Nutrient Digestibility 

Over the measurement period of 45 days, fecal output was collected and quantified from each sheep twice a day before each meal. A metal mesh was installed under each pen to allow urine to drain through while collecting feces through an inclined net. The weekly feces were bulked for each sheep, subsampled and oven-dried at 65 °C for 48 h to quantify the DM content. Dried feces were ground through a 1 mm sieve (Cyclotec 1093; FOSS Ltd., Hillerød,) and analyzed for gross energy, N, NDF and ADF, using the same methods as described above for feed analysis.

### 2.5. Average Daily Gain and Feed Conversion Efficiency

The live weights were measured before feeding in the morning on the first and final days of the measurement period. The ADG and feed conversion efficiency (FCE) were calculated for each sheep over the 45 days of the measurement period.

### 2.6. Rumen Sampling

A rumen fluid sample (~20 mL) was taken via a rumen fistula every two hours for 24 h on day 45 of the measurement period and filtered through two layers of cheesecloth. The pH of the filtered rumen fluid was measured immediately using a standard pH meter (PHM210; Radiometer Analytical, Copenhagen, Denmark). A subsample (10 mL) was centrifuged at 20,000× *g* for 5 min at 4 °C, and the supernatants were then transferred into new tubes and frozen at −20 °C until analysis. For VFA analysis, 100 μL of the rumen supernatant was placed into a 2 mL Eppendorf tube, and 20 μL of the internal standard and 40 μL of metaphosphoric acid were added. The samples were then diluted 10 times with a 50:50 acetone/water diluent. The concentrations of acetate, propionate and butyrate were then determined as described by Chen and Lifschitz (1989) using a Gas Chromatographer (GC: Shimadzu GC-2010, Japan) fitted with an SGE BP21 30 m × 530 μm × 1.0 μm wide-bore capillary column [16]. The GC column temperature was held at 105 °C for 4 min, then ramped at 15 °C/min to 200 °C and then subjected to bake off at 230 °C for 5 min. The injector temperature was 240 °C, the Flame Ionization Detector (FID) temperature was 240 °C, H_2_ pressure was 34.2 kPa, the column flow was 5.23 mL/min with 1:30 split ratio and N_2_ purge flow was 3.0 mL/min.

### 2.7. Blood Sampling 

Blood samples were collected from the vena jugularis before feeding at 8:00 a.m. on days 30 and 45 of the measurement period and then centrifuged at 1000× *g* for 5 min at room temperature. Serum was collected and stored at 20 °C until analysis. Serum samples were thawed at 4 °C, bulked per sheep and analyzed using an Automatic Biochemical Analyzer (AU480; Beckman Coulter, Brea, CA, USA) for total protein (TP), aspartate aminotransferase (AST), alanine aminotransferase (ALT), immunoglobulin A(IgA), immunoglobulin G(IgG), total antioxidant capacity (T-AOC) and superoxide dismutase (SOD) using ELISA Kits (Nanjing Jiancheng Bioengineering Institute, Nanjing, China) according to the instructions.

### 2.8. Statistical Analysis

GenStat 16 statistical software was used for statistical analysis. Initial live weight on day 0 of the measurement period was included as a covariate for the analysis of ADG and FCE using one-way ANOVA. Daily DMI, blood and feces measurements were averaged across the measurement days for each sheep, and the means were analyzed using one-way ANOVA. Rumen measurements at each sampling time were analyzed by repeated measurements in the animals as replicates. The significance of the treatment effect was declared when *p* < 0.05.

## 3. Results

### 3.1. Rumen pH and Volatile Fatty Acid Concentration

A significant interaction between treatment and time of sampling affected ruminal pH (*p* < 0.001; Table 1), which was higher for the supplemented than for the control group at 2:00 a.m., 2:00 p.m., 4:00 p.m. and 6:00 p.m., but lower at 6:00 a.m., 10:00 a.m., 10:00 p.m. and 12:00 a.m. (Figure 1). The interaction treatment–time of sampling was also significant for butyrate concentration (*p* < 0.001), which was higher for the supplemented than for the control group at 2:00 a.m., 4:00 a.m., 10:00 a.m., 12:00 p.m, 2:00 p.m., 4:00 p.m. and 8:00 p.m., but lower at 10:00 p.m. and 12:00 a.m. (Figure 2). No treatment–time of sampling interactions were significant for the concentrations of acetate and total VFA, and for the acetate/propionate ratio, which were all lower for the supplemented group than the control group.

### 3.2. Blood Indices

No significant differences were found in TP, AST and ALT between the two groups (*p* > 0.05), but both IgA and IgG were higher in the supplemented group than in the control group (*p* < 0.001; Table 2). The T-AOC and SOD in serum were higher (*p* < 0.001) in the supplemented than in the control group as well.

### 3.3. Chemical Composition, Intake, Average Daily Gain and Apparent Nutrient Digestibility

The basal diet content of CP, NDF, and ADF were 10.7%, 65.9% and 49.6%, respectively, on DM basis. The content of glycyrrhizin, licoflavone, ash, carbohydrate in the licorice root extract were 22%, 40%, 13% and 25%, respectively, on DM basis. These values were not measured in this study and were supplied by the manufacturer. Daily DMI was higher (*p* < 0.05) in the control than in the supplemented group, whereas apparent nutrient digestibility, ADG and FCE did not differ between the two groups (Table 3). 

## 4. Discussion 

### 4.1. Rumen Fermentation Characteristics

The mean rumen pH did not differ between treatments in the current study, averaged at 6.6 and ranged between 6.1 and 6.8 across both groups. These values are within the normal range reported previously for sheep fed a cotton seed hull-based diet (6.0–7.0) [17], and a pH above 6.0 would not limit rumen microbial function [18].

A licorice root extract supplementation reduced total VFA and acetate productions and increased propionate and butyrate productions compared to the control. Over a 24 h period, the supplemented group had a consistently higher butyrate content than the control group. This may be attributed to the high content of glycyrrhizin (22% of DM) in the licorice root extract fed to sheep. Similarly, Lila et al. (2005) reported a reduction in total VFA and acetate concentrations and an increase in propionate and butyrate concentrations in steers supplemented with saponin at 1.0% of DM [19]. Further, the concentrations of acetate and total VFA decreased, and the concentration of propionate increased in goats supplemented with 26 mg of saponin/kg body weight from *Biophtum petersianum* [20] (0.1% on feed DM basis). Saponin is known to inhibit the growth of acetate-producing bacteria (e.g., *Butyrivibrio fibrisolvens*) and reduce the population of protozoa by reacting with cholesterol in the membrane of protozoa in the rumen [21], causing protozoa to rupture [22]. Protozoa elimination through defaunation can reduce the ratio of acetate to propionate [22,23,24]. This may be one of the reasons why a 4.5% licorice root extract supplementation reduced the ratio of acetate to propionate in the present study. The fluctuation of pH may reflect an interaction between feed intake, urea recycling from magnesium and acid production in the rumen [25] (Wang et al., 2017). It is worth noting that in the present study, the rumen liquor was sampled for one single day, and the results should be treated with caution. 

### 4.2. Blood Indices

In the current study, there were no differences observed in TP, AST and ALT between the two groups. Similar results were reported in serum AST and ALT when a 0.5% licorice root extract was supplemented [10]. Immunoglobulins provide an indication of immune system status and are proteins that help in eliminating foreign agents from the body, such as bacteria and viruses. In the current study, licorice root extract supplementation increased IgA and IgG levels. This is in agreement with the results of Sun and Pan [26], showing that IgA and IgG levels increased when licorice saponin (75.7% *G. uralensis* saponin content) was added to a mouse diet. Furthermore, Sajjadi et al. (2014) showed that a 0.56% licorice (no saponin content reported) supplementation increased the immune response of female Holstein calves [9]. The higher IgA and IgG levels in the supplemented group compared with the control group in this study imply that licorice root extracts might enhance the immune status in Karakul sheep.

Licorice extracts have been known to improve the antioxidant capacity of animals [27]. T-AOC and SOD are commonly used as indicators of the antioxidant status of animals. In addition, the antioxidant enzymes SOD and glutathione (GSH) play a fundamental role in cellular defence against reactive free radicals and other oxidant agents [28]. The current study showed higher T-AOC and SOD levels in the supplemented group compared with the control group. This result is in line with the findings of Zhang et al. (2015), who showed that supplementation of a 0.4% licorice root extract (16.4% total flavonoids content) to sheep diet resulted in a greater antioxidant capacity of meat compared with the non-supplemented group [8]. Such enhanced antioxidant capacity was likely due to the higher content of licorice flavonoids in the licorice extract [29]. Overall, our results indicate a potential improvement of the antioxidant status of sheep supplemented with a 4.5% licorice root extract.

### 4.3. Animal Performance and Apparent Nutrient Digestibility 

In the present study, a licorice root extract supplemented to a cottonseed hull-based diet for Karakul sheep resulted in a decreased DMI. The reason behind this is unknown. A recent study showed no change in DMI of Hanwoo steers supplemented with 0.5% licorice compared with that of un-supplemented steers [10]. Similarly, a study on heifer calves showed no difference in DMI when the animals were supplemented with 10 g day^−1^ licorice (~0.56% of the diet DM), compared with the un-supplemented group [9]. However, a number of studies showed that a high level of plant extract supplementation can affect feed palatability and consequently alter DMI [30,31]. Therefore, we speculate that the reduction of DMI in the current study for the supplemented group may be related to the high level of licorice root extract (4.5%) present in the diet. Further study is needed to validate this speculation.

Despite the reduction in DMI in the licorice root extract-supplemented group, no difference in ADG was observed between the two groups in this study. It is important to note that future work to investigate ADG should use young sheep in their growing phase. A higher number of sheep would also provide more power for the ADG analyses. Further, a 4.5% licorice root extract supplementation (glycyrrhizin content equivalent to 1.0% DM) had limited effects on the apparent digestibility of CP, NDF and ADF in the present study. This is in agreement with results of other plant extracts enriched in saponin supplemented to the diet of ruminants [20,32,33].

## 5. Conclusions

Supplementation of a 4.5% licorice root extract (i.e., 1.0% glycyrrhizin of total DM) to the diet of Karakul sheep reduced DMI without changing ADG, FCE, apparent nutrient digestibility and rumen fermentation and improved the immune and antioxidant status in sheep. Future studies are needed to confirm the effects of individual bioactive chemical compounds contained in licorice on sheep, as limited ruminant studies in the literature have reported the actual chemical composition of licorice or licorice extracts, making it difficult to compare different published studies.

## Figures and Tables

**Figure 1 animals-09-00279-f001:**
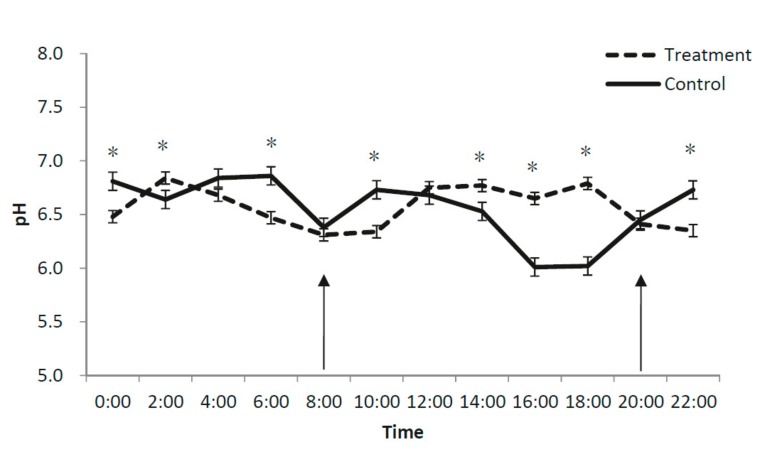
Diurnal effects of a 4.5% licorice root extract supplementation on rumen pH of Karakul sheep fed cottonseed hull-based diets. Arrows: feeding time; Bar: SEM; *, significant difference at *p* < 0.05.

**Figure 2 animals-09-00279-f002:**
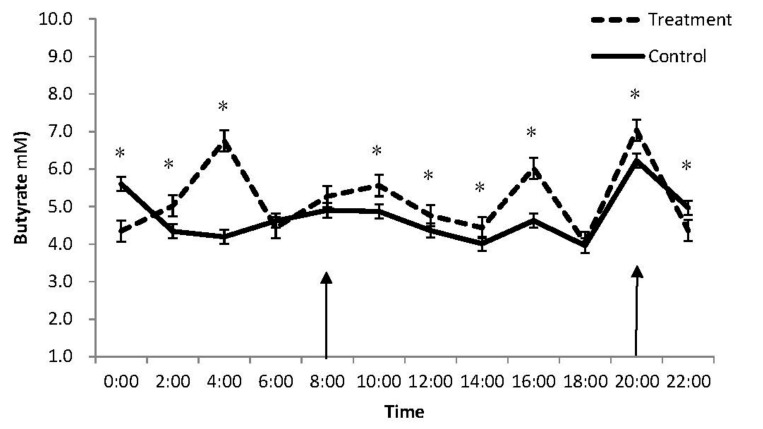
Diurnal effects of a 4.5% licorice root extract supplementation on rumen butyrate concentration of Karakul sheep fed cottonseed hull-based diets. Arrows: feeding time; Bar: SEM; *, significant difference at *p* < 0.05.

**Table 1 animals-09-00279-t001:** Effects of a 4.5% licorice root extract supplementation on the daily means of ruminal pH and volatile fatty acid concentration of Karakul sheep fed cottonseed hull-based diets.

	Control	Supplemented	SEM		*p-*Value	
				Treatment	Time	Treatment–Time
Rumen pH	6.6	6.6	0.01	0.537	<0.001	<0.001
Acetate (mM)	31.4	27.8	0.39	<0.001	0.002	0.597
Propionate (mM)	13.0	14.2	0.21	0.002	<0.001	0.130
Butyrate (mM)	4.7	5.2	0.07	0.038	<0.001	0.011
Total volatile fatty acid (mM)	49.1	47.2	0.49	0.019	<0.001	0.424
Acetate: propionate ratio	2.52	1.96	0.17	<0.001	0.302	0.422

SEM: Standard error of mean. Total volatile fatty acid: Acetate + Propionate + Butyrate.

**Table 2 animals-09-00279-t002:** Effects of a 4.5% licorice root extract supplementation on blood indices of Karakul sheep fed cottonseed hull-based diets.

	Control	Supplemented	SEM	*p*-Value
Total protein (g L^−1^)	86.4	86.8	2.29	0.905
Aspartate aminotransferase (U L^−1^)	131.1	131.7	11.62	0.803
Alanine aminotransferase (U L^−1^)	16.2	16.1	0.41	0.333
Immunoglobulin A (ng mL^−1^)	4.3	9.1	0.12	<0.001
Immunoglobulin G (ng mL^−1^)	44.6	82.2	1.05	<0.001
Total antioxidant capacity (U mL^−1^)	7.3	12.9	0.34	<0.001
Total superoxide dismutase (U mL^−1^)	87.5	105.5	1.61	<0.001

**Table 3 animals-09-00279-t003:** Effects of a 4.5% licorice root extract supplementation on intake, average daily gain, feed conversion efficiency and apparent nutrient digestibility of Karakul sheep fed cottonseed hull-based diets.

	Control	Supplemented	SEM	*p-*Value
Initial live weight (kg)	30.43	30.38	0.243	0.739
Dry matter intake (kg day^−1^)	0.89	0.85	0.011	0.013
Digestible energy intake (MJ day^−1^)	8.4	8.2	0.09	0.176
Average daily gain (g day^−1^)	86.7	71.1	21.77	0.702
Feed conversion efficiency (g kg^−1^)	93.3	86.2	26.08	0.878
**Apparent Nutrient Digestibility (%)**				
Dry matter	48.9	50.9	0.69	0.067
Crude protein	47.5	49.5	0.74	0.092
Neutral detergent fiber	39.8	42.1	0.76	0.054
Acid detergent fiber	37.3	39.8	0.87	0.073

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
