# Peer review of "Effects of Licorice Extract Supplementation on Feed Intake, Digestion, Rumen Function, Blood Indices and Live Weight Gain of Karakul Sheep"

_animals, 2019, doi:10.3390/ani9050279_

Round 1

Reviewer 1 Report

General comments:

This study investigated the effects of dietary supplementation with licorice extract on growth, rumen function and blood indices of fistulated 1.5 years old wethers.

The aim of the study is clearly stated, introduction is consistent with the aims, and in general the paper is well written. Methodologies used are generally consistent with the aims.

My main remarks/questions concern:

- I am not convinced that this study, based on 12 1.5 years old wethers, can provide sound information about the effects of licorice extract on growth rate. This can be better do with a study based on more animals in their growing/finishing phase. Growth rate and feed conversion have been measured and can be presented in tables, but I would limit the aims of the study to rumen functions and blood indices, and would limit the discussion and emphasis about the effects on AGD (L 200 – 204; L 227 – 230);

- as authors are specifically interested on blood indices, and have taken blood samples on days 30 and 45 of trial, I ask why they have not considered to sample animals also at the starting of the trial, for investigating the trend of variation of blood indices, and possible differences between control and supplemented wethers;

- in any case, as blood indices have been determined in 2 different periods, they could be better analyzed as repeated observations rather than as averages of 2 days (they are not replicates, but measures made in different periods).

Below you can find some specific comments.

Simple summary

I understand that simple summary must be concise, but this one is in my opinion too concise. Simple summary should state the problem addressed, the aims and objectives, main results and conclusions, and possibly implications. I think it should be improved.

L 17 no abbreviation in simple summary

Material and methods

Some information in the Animal ethical code section are redundant with experimental design (see for instance L 80 – 81 and L 102-104; L 82-83 and L 101-102; L 89-92 and L 104-106).  Try to rearrange this first section of material and methods.

L 79-80. 2009 and 2010? Are you sure? These data are 10 years old?

L 82: average initial liveweight here (28 kg) not consistent with that reported in table 1 (30.4 kg)

L 158 Why have you inserted initial live weight as covariate in the analysis of ADG ad FCE? Initial LW is not different between treatments. Is the covariate significant?

L 161 I think that blood indices should be analyzed as repeated observations rather than as averages of 2 days. You could run an interaction period x treatment.

Results

L 170 the amount of glycyrrhizin is already given in M&M (L 98)

L 171 and table 1 licorice supplemented animals showed significantly lower DMI (- 4.5%, P = 0.013) but only slightly lower digestible energy intake (- 2%, not significant difference). Had the diets the some energy content or did licorice supplementation add some extra energy to the diet? Or, otherwise, comparable energy intake despite lower DMI is due to greater digestibility?  You have not discussed these results  

Discussion

L 200 – 204 I still believe that 1.5 years old wethers  are not the best category for investigate the effects of dietary treatments on growth rate. Moreover, average nominal difference between groups was close to 20%; as it was not significant, I think that variation of ADG was high and/or power of your study for ADG was limited due to reduced sample size. For these reasons I would be cautious about results on ADG and FCE

L 204- 207 apparent digestibility of DM and CP was tendentially improved. I don’t understand what was your hypothesis/expectations about effects on digestibility. Maybe this point could deserve a bit more discussion.    

Author Response

Dear Reviewers and editor,

Thanks for your constructive comments.

Please see our response to your comments below and changes made in color in the manuscript.

Open Review

English language and style

( ) Extensive editing of English language and style required 
( ) Moderate English changes required 
(x) English language and style are fine/minor spell check required 
( ) I don't feel qualified to judge about the English language and style 

Yes

Can be improved

Must be improved

Not applicable

Does the introduction provide sufficient background and include all relevant references?

(x)

( )

( )

( )

Is the research design appropriate?

(x)

( )

( )

( )

Are the methods adequately described?

( )

(x)

( )

( )

Are the results clearly presented?

( )

(x)

( )

( )

Are the conclusions supported by the results?

(x)

( )

( )

( )

Comments and Suggestions for Authors

General comments:

This study investigated the effects of dietary supplementation with licorice extract on growth, rumen function and blood indices of fistulated 1.5 years old wethers.

The aim of the study is clearly stated, introduction is consistent with the aims, and in general the paper is well written. Methodologies used are generally consistent with the aims.

Thanks for the above positive comments.

My main remarks/questions concern:

- I am not convinced that this study, based on 12 1.5 years old wethers, can provide sound information about the effects of licorice extract on growth rate. This can be better do with a study based on more animals in their growing/finishing phase. Growth rate and feed conversion have been measured and can be presented in tables, but I would limit the aims of the study to rumen functions and blood indices, and would limit the discussion and emphasis about the effects on AGD (L 200 – 204; L 227 – 230);

We agree that younger sheep/more sheep may be a better model to study ADG. However, 45 days ADG measurement should have provided long enough time to quantify ADG in this study.

Taking into reviewer comment on board, we reduced the discussion of growth performance and feed conversion in this manuscript. We added in a sentence to suggest younger sheep/more sheep is needed for future study. Please see line 246-248.

- as authors are specifically interested on blood indices, and have taken blood samples on days 30 and 45 of trial, I ask why they have not considered to sample animals also at the starting of the trial, for investigating the trend of variation of blood indices, and possible differences between control and supplemented wethers;

We did not aim to investigate the blood indices trend in this study. Literature suggested that 30 days is long enough for those blood indices we measured to change in response to dietary treatment. If we were running a trial for 10 days, we would have taken day 0 blood as a baseline and include it as a co variate in analysis.

- in any case, as blood indices have been determined in 2 different periods, they could be better analyzed as repeated observations rather than as averages of 2 days (they are not replicates, but measures made in different periods).

Due to the limited budget, we combined the serum of 30d and 45d into one sample per sheep for analysis. Please see line 151-152.

Below you can find some specific comments.

Simple summary

I understand that simple summary must be concise, but this one is in my opinion too concise. Simple summary should state the problem addressed, the aims and objectives, main results and conclusions, and possibly implications. I think it should be improved.

Revised the simple summary. Please see line 16-21.

L 17 no abbreviation in simple summary

Revised. Please see line 19. 

Material and methods

Some information in the Animal ethical code section are redundant with experimental design (see for instance L 80 – 81 and L 102-104; L 82-83 and L 101-102; L 89-92 and L 104-106).  Try to rearrange this first section of material and methods.

We have re-written this part according to the Reviewers suggestion. Please see line 102-105.

L 79-80. 2009 and 2010? Are you sure? These data are 10 years old?

Yes, it is.

L 82: average initial liveweight here (28 kg) not consistent with that reported in table 1 (30.4 kg)

I am sorry for that weight = 28 kg was a mistake, 29.4 kg was the weight before the adaptation, and 30.4 kg was the weight before the measurement period. We have revised it. Please see line 85.

L 158 Why have you inserted initial live weight as covariate in the analysis of ADG ad FCE? Initial LW is not different between treatments. Is the covariate significant?

The standard deviation of the initial body weight was relatively large, and considering that the initial body weight might have an impact on daily gain and feed conversion efficiency, the initial body weight was taken as a covariance factor for statistical analysis, though the covariance statistics showed no significant difference.

L 161 I think that blood indices should be analyzed as repeated observations rather than as averages of 2 days. You could run an interaction period x treatment.

Sorry, we were not interested in the period effect. Blood samples from 2 periods were bulked per sheep prior analysis. 

Results

L 170 the amount of glycyrrhizin is already given in M&M (L 98)

We have deleted it.

L 171 and table 1 licorice supplemented animals showed significantly lower DMI (- 4.5%, P = 0.013) but only slightly lower digestible energy intake (- 2%, not significant difference). Had the diets the some energy content or did licorice supplementation add some extra energy to the diet? Or, otherwise, comparable energy intake despite lower DMI is due to greater digestibility?  You have not discussed these results  

No additional energy substances were added to the diet, although the digestibility was not identical between groups, the difference was not significant. Digestible energy intake reflected both DMI and digestible energy content measured from animal in vivo.

Discussion

L 200 – 204 I still believe that 1.5 years old wethers  are not the best category for investigate the effects of dietary treatments on growth rate. Moreover, average nominal difference between groups was close to 20%; as it was not significant, I think that variation of ADG was high and/or power of your study for ADG was limited due to reduced sample size. For these reasons I would be cautious about results on ADG and FCE

We agree that younger sheep/more sheep may be a better model to study ADG. However, 45 days ADG measurement should have provided long enough time to quantify ADG in this study.

Taking into reviewer comment on board, we reduced the discussion of growth performance and feed conversion in this manuscript. We focused this study on blood and rumen parameters.

L 204- 207 apparent digestibility of DM and CP was tendentially improved. I don’t understand what was your hypothesis/expectations about effects on digestibility. Maybe this point could deserve a bit more discussion.     

Though there is a trend, the change is small (~4%). We dont think it would impact on production. Therefore, no discussion is added in. Future trial with more animal will need to explore this area. Please see line 247-248.

Submission Date

01 April 2019

Date of this review

19 Apr 2019 16:27:52

Best regards,

Dr. Guo Xuefeng behalf all the authors

Reviewer 2 Report

Xue-Feng Guo et al. – Manuscript submitted to ANIMALS

 “Effects of licorice extract supplementation on feed intake, digestibility, live weight gain, rumen function and blood indices of Karakul sheep

The paper aims at evaluating the effects of licorice extract supplementation on performance, ruminal functions and selected blood indices (mainly those referring to the oxidative metabolism) in sheep. The experimental design, aim, and methods are clearly described.

However, I have a few concerns about a number of minor items:

Lie 35 – “has” instead of “have”

Line 47 – I believe Fisch should be not in italic.

Line 49 – I would use “scarcely” or “little” instead of “few”.

Line 69 – “at 4.5% of DM”: is it a repetition within the same sentence?

Line 74 – “based” instead of “base”?

Line 86 – “vitamin premix” instead of “vitamins premix” is more common.

Lines 104-106 – this is a repetition of what already described in Materials and Methods. Is it useful to report this sentence again in this section?

Line 125-126 – how did the author check that urine was not contaminated by faeces? Was there any technical solution, in the metabolic cage design, able to prevent this contamination?

Lines 131-133 – Usually live weight, especially in ruminants, has to be measured in two consecutive days. In this trial, where differences in live weight were evaluated, this should have been taken into account. I believe that both the limited number of animals and the way live weight was recorded, should be mentioned in the discussion (when discussing about growth performance).

Line 157 – If tendencies were taken into account (see line 172), this should be mentioned in the Statistical Analysis section (es. “tendencies were considered for P<…)

Line 194 – “showed” instead of “showen”.

Lines 210-211 – please discuss the interaction time of sampling x treatment for both rumen pH and butyrate.

Line 226 – please check the term “reduced”.

Line 230-231 – it is almost inexplicable the reason why the author did not sample the rumen contents more often during the experiment. Having fistulated animals in the trial without using them does not have a clear explanation. More rumen samples would allow for a greater amount of data; this would have helped in increasing the number of observations in the statistical analysis. I think an explanation for this choice would be appropriate.

Lines 240-243- A tentative explanation by the authors would surely be appreciated, based on the existing literature on antioxidant compounds used in animal nutrition.

Line 257 - The variability of growth parameters, with only six animals per treatment, is much too high to allow the detection of significant differences between means. Probably a higher number of animals (not necessarily with ruminal fistula) would be appropriate to measure changes in live weight and FCE. This aspect should be considered in the discussion and/or the conclusion.

Tables 1 and 3 – please check the alignment of the rows.

Author Response

Dear Reviewers and editor,

Thanks for your constructive comments.

Please see our response to your comments below and changes made in color in the manuscript.

Open Review

English language and style

( ) Extensive editing of English language and style required 
( ) Moderate English changes required 
(x) English language and style are fine/minor spell check required 
( ) I don't feel qualified to judge about the English language and style 

Yes

Can be improved

Must be improved

Not applicable

Does the introduction provide sufficient background and include all relevant references?

(x)

( )

( )

( )

Is the research design appropriate?

( )

(x)

( )

( )

Are the methods adequately described?

( )

(x)

( )

( )

Are the results clearly presented?

(x)

( )

( )

( )

Are the conclusions supported by the results?

(x)

( )

( )

( )

Comments and Suggestions for Authors

Xue-Feng Guo et al. – Manuscript submitted to ANIMALS

 “Effects of licorice extract supplementation on feed intake, digestibility, live weight gain, rumen function and blood indices of Karakul sheep

The paper aims at evaluating the effects of licorice extract supplementation on performance, ruminal functions and selected blood indices (mainly those referring to the oxidative metabolism) in sheep. The experimental design, aim, and methods are clearly described.

Thanks for the positive comments.

However, I have a few concerns about a number of minor items:

Lie 35 – “has” instead of “have”

Corrected. Please see line 37.

Line 47 – I believe Fisch should be not in italic.

Deleted it.

Line 49 – I would use “scarcely” or “little” instead of “few”.

Corrected. Please see line 52.

Line 69 – “at 4.5% of DM”: is it a repetition within the same sentence?

Deleted it.

Line 74 – “based” instead of “base”?

Corrected. Please see line 78.

Line 86 – “vitamin premix” instead of “vitamins premix” is more common.

Corrected. Please see line 88.

Lines 104-106 – this is a repetition of what already described in Materials and Methods. Is it useful to report this sentence again in this section?

We have re-written this part according to the Reviewers suggestion. Please see line 102-105.

Line 125-126 – how did the author check that urine was not contaminated by faeces? Was there any technical solution, in the metabolic cage design, able to prevent this contamination?

We used the metabolism cage with the pore size of feed sieve which is larger than the fecal particles. Urine can be leaked into the collection bag through a inclined plane, and the urine is directed into the collection bucket, ensuring that the fecal and urine are separated and not contaminated. Please see line 124-125.

Lines 131-133 – Usually live weight, especially in ruminants, has to be measured in two consecutive days. In this trial, where differences in live weight were evaluated, this should have been taken into account. I believe that both the limited number of animals and the way live weight was recorded, should be mentioned in the discussion (when discussing about growth performance).

We agree that younger sheep/more sheep may be a better model to study ADG. However, 45 days ADG measurement should have provided long enough time to quantify ADG in this study.

Taking into reviewer comment on board, we reduced the discussion of growth performance and feed conversion in this manuscript. We focused this study on blood and rumen parameters.

Line 157 – If tendencies were taken into account (see line 172), this should be mentioned in the Statistical Analysis section (es. “tendencies were considered for P<…)

We have deleted the description and discussion of results, the difference is small between treatments (~4%).

Line 194 – “showed” instead of “showen”.

Corrected. Please see line 236.

Lines 210-211 – please discuss the interaction time of sampling x treatment for both rumen pH and butyrate.

The discussion section is revised. Please see line 193-194 and 206-209.

Line 226 – please check the term “reduced”. 

We have check it.

Line 230-231 – it is almost inexplicable the reason why the author did not sample the rumen contents more often during the experiment. Having fistulated animals in the trial without using them does not have a clear explanation. More rumen samples would allow for a greater amount of data; this would have helped in increasing the number of observations in the statistical analysis. I think an explanation for this choice would be appropriate.

We agree that rumen sample more often will benefit the analysis. However, due to animal welfare concern and limited my budget, we did not sample rumen more frequently in this study.

Lines 240-243- A tentative explanation by the authors would surely be appreciated, based on the existing literature on antioxidant compounds used in animal nutrition.

We discussed the antioxidant compounds of Licorice extract in line 222-232.

Line 257 - The variability of growth parameters, with only six animals per treatment, is much too high to allow the detection of significant differences between means. Probably a higher number of animals (not necessarily with ruminal fistula) would be appropriate to measure changes in live weight and FCE. This aspect should be considered in the discussion and/or the conclusion.

We agree that younger sheep/more sheep may be a better model to study ADG. However, 45 days ADG measurement should have provided long enough time to quantify ADG in this study.

Taking into reviewer comment on board, we reduced the discussion of growth performance and feed conversion in this manuscript. We focused this study on blood and rumen parameters.

Tables 1 and 3 – please check the alignment of the rows.

We have checked the it. Please see table 2 and table 3.

Submission Date

01 April 2019

Date of this review

09 May 2019 09:06:03

Best regards,

Dr. Guo Xuefeng behalf all the authors
